# Holographic transcranial ultrasound neuromodulation enhances stimulation efficacy by cooperatively recruiting distributed brain circuits

Hector Estrada [1,2], Yiming Chen [1,2,7], Théo Lemaire [3,4], Neda Davoudi [1,2], Ali Özbek[1,2], Qendresa Parduzi [1,2,5], Shy Shoham [3,4,6] ✉ & Daniel Razansky [1,2] ✉

Precision-targeted ultrasonic neuromodulation offers immense potential for studying brain function and treating neurological diseases. Yet, its application has been limited by challenges in achieving precise spatio-temporal control and monitoring of ultrasound effects on brain circuits. Here we show that transcranial ultrasound elicits direct and highly focal responses, which can be dynamically steered at spatio-temporal scales relevant for neural function. Furthermore, holographic transcranial ultrasound stimulation allows direct control of the stimulated volume and actively modulates local and mid-range network projections, effectively lowering the activation threshold by an order of magnitude. To better understand this previously unexplored excitability regime not fully explained by the conventional pressure–frequency dyad, we developed a dual modelling framework, where both an empirical and a mechanistic model were constructed to capture the intricacies of holographic transcranial ultrasound stimulation. These models achieve qualitative agreement with our experimental results, suggesting that these findings are predominantly driven by putative network interactions. Our results bring insight on the complex interaction mechanisms of ultrasound with neural tissue and highlight its potential for the noninvasive interfacing of distributed brain networks.

Efforts to modulate brain function are instrumental towards tackling neurological diseases in an aging population[1,2] and for advancing basic neuroscience research[3,4]. Transcranial ultrasound stimulation (TUS) stands out as a promising technique[5,6] for non-invasively modulating brain function with orders-of-magnitude greater precision and depth penetration than its electromagnetic neurostimulation counterparts[7]. Single-element ultrasound (US) transducers have been the main tool for studying ultrasound neuromodulation effects in various species' brains[8–15] due to their experimental simplicity and affordability, while US arrays have facilitated simultaneous neuromodulation and neuroimaging[16,17], offering a way to compensate for skull-induced aberrations and to perform electronic steering.

To fully harness TUS' potential for exploring distributed brain network functions, several knowledge gaps must be addressed. First, neural responses to TUS depend on a multi-dimensional parameter space, including pressure, intensity, spatial parameters (like focal dimensions) and temporal parameters (for example, US frequency, pulse duration, pulse repetition frequency and duty cycle[6]). Despite extensive

**Fig. 1 | Noninvasive, precise and highly controllable delivery of ultrasound perturbations to the mouse brain. a**, The integrated stimulation-monitoring system uses a 512-element spherically focused array for hTUS delivery (blue) navigated with volumetric optoacoustic tomography (red) performed with the same array. Widefield fluorescence calcium imaging (cyan) is performed simultaneously using a fibrescopic insert integrated into a central aperture of the array. CW, continuous wave; PC, personal computer. **b**, Schematic of hTUS with random array subsets rendered in different colours. **c**, hTUS characterization with hydrophone and numerical simulation (top), and minimum intensity projection of the in vivo transcranial FTT (bottom) near the somatosensory area. **d**, Ultrasound focus characterization from data in **c**. FWHM, full width at half maximum. **e**, Transcranial ultrasound pressure of the 3 MHz wave measured with a hydrophone.

parametric testing in vitro[18,19] and in vivo[9,10,20], a comprehensive framework is still lacking, mainly due to the lack of direct measurements of US-evoked neural activity. Other key early attempts to directly measure TUS-induced responses were contaminated by non-specific auditory responses[21] or thermal confounds[22] or otherwise aimed at characterizing indirect measures, such as evoked muscle electromyographies[9,23] or haemodynamic responses[24,25]. Moreover, precise application of this tool faces physical constraints that limit in vivo observations and their interpretation. For example, single-element transducers produce different focal sizes at different frequencies due to wave diffraction. Thus, any attempt to evaluate the effect of frequency on stimulation with these transducers is inevitably confounded by the varying size of the stimulated area[26]. Also, the lower (clinically relevant) frequencies used to traverse the human skull (<1 MHz) typically stimulate excessive brain volumes in small rodents[27]. Finally, previous studies have primarily focused on single-spot TUS while disregarding the contribution of network interactions to US neuromodulation and the inherent interplay between connectivity and evoked effects.

To overcome these challenges, we introduce a combination of technical and theoretical advances. We use a spherical matrix US array[16] to generate holographically distributed TUS (hTUS) cortical stimulation in mice, achieving either single or multiple adjacently grouped high-resolution foci. This approach effectively decouples the coverage area from the dimensions and frequency of single-spot stimulation, offering a method to investigate network recruitment. Our system also enables direct observation of localized evoked responses by applying a computational compensation strategy[28] to a fully integrated concurrent widefield fluorescence imaging. We use this integrated solution to compare hTUS calcium activation dynamics against single-spot TUS results at the same frequency and find a large shift in the threshold excitation pressure that cannot be explained using existing single-neuron-level hypotheses. These findings may also help to reconcile multiple contradicting observations where network and neuron-level responses observed in vivo may have been confounded and misinterpreted.

## Results

### Precise TUS and hTUS delivery and simultaneous Ca²⁺ imaging
We delivered focal US by a custom-made 512-channel matrix array (Fig. 1a), conforming to a spherical surface with a 150° (ref. 29) coverage. Unlike single-element transducers, the phased array achieves a precise focus (~250 μm laterally, ~560 μm axially at our working frequency of 3 MHz (Fig. 1e), which can also be steered in three dimensions[16] by adjusting relative delays of the driving signals (Fig. 1b). Using the principle of reciprocity, this array can also be used in reception mode to precisely localize the stimulation spot in the mouse brain with optoacoustic tomography[16]. In addition, it is capable of generating multiple hTUS foci simultaneously offering exceptional control over the pressure field and the targeted brain region (Supplementary Figs. 1–3). Consistent with free-field measurements and simulations (Fig. 1c, top), hTUS maintains the holographic pattern through the mouse skull, as demonstrated in vivo (Fig. 1c, bottom) using the fluorothermal tag (FTT; Methods)[29]. Skull-induced distortions are relatively minor, namely, a weak nonuniformity in focus intensity caused by the skull's curvature (that is, the two foci on the left versus the focus on the right in Fig. 1c) and a slight (10–30%) broadening of the focus observed via fluorothermal dip measurements in vivo (Fig. 1d), which could partially be attributed to the effect of thermal diffusion.

### TUS induces localized cortical activation
To explore the cortical responses dynamics to precise TUS, we performed concurrent rapid (20 frames per second (f.p.s.)) widefield fluorescence imaging through a fibrescope in anaesthetized Thy1 transgenic mice expressing the calcium (Ca²⁺) indicator GCamp6f in cortical layers[30]. The right hemisphere was sonicated 20 times at 10 s intervals with continuous 150 ms US pulses ($20 < I_{SPPA} < 131$ W cm⁻²; $I_{SPPA}$

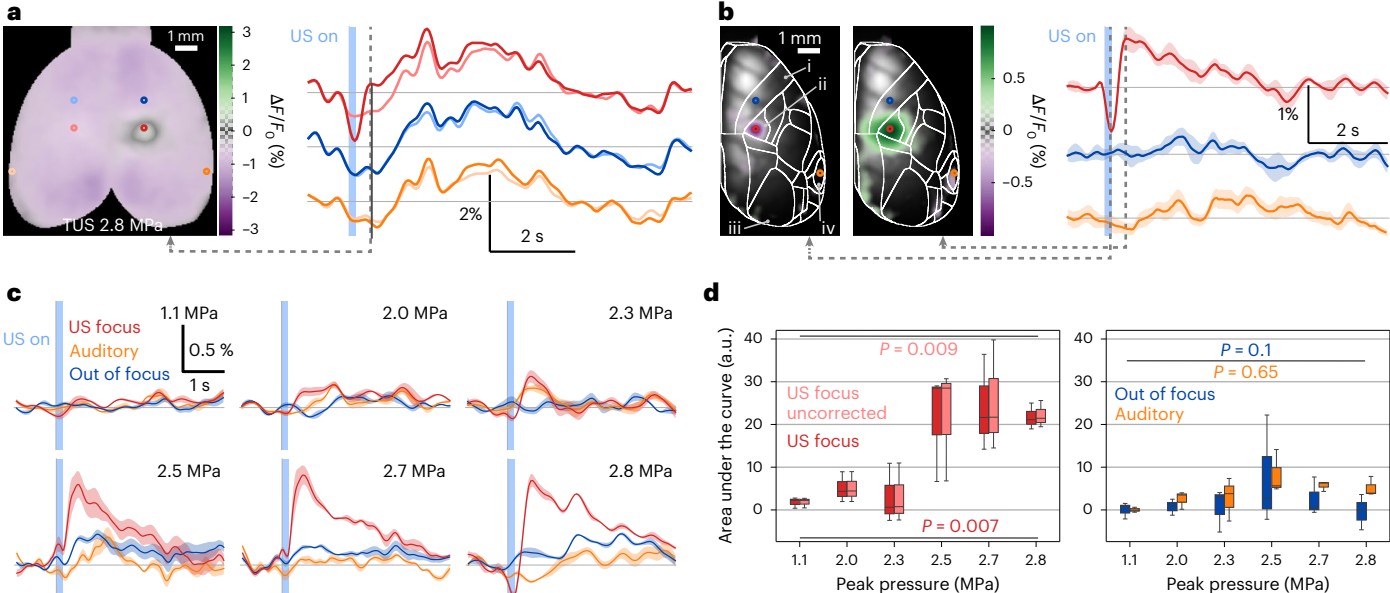

**Fig. 2 | TUS induces localized neural responses. a**, Snapshot of calcium changes across the mouse cortex and time traces at selected points. The instant of the snapshot is indicated by a vertical dashed line on the time traces. US pulse duration is represented as light blue region. The sonicated point is red (1.2 mm posterior to bregma, 1.6 right to midline), the out-of-focus is blue (1.2 mm anterior to focus point) and the auditory cortex is orange. **b**, Resting-state-corrected snapshots. Spatial maps are overlaid with a standard mouse brain atlas to facilitate the identification of cortical regions. Time traces shown as mean ± s.e.m. Anatomical labels: i, primary motor area; ii, primary somatosensory area (lower limb); iii, primary visual area; iv, primary auditory area. **c**, Progression of the $Ca^{2+}$ signal with increasing peak pressure using FTT correction plotted as mean ± s.e.m ($n$ = 3, average of ~20 stimulations each). **d**, Change in the area under the time trace at the stimulated spot as a function of the peak pressure between 0.45 and 2 s from the US sonication start. Whiskers extend from minimum to maximum, the box from quartiles 25th to 75th, while the horizontal line shows the median. One-way ANOVA ($n$ = 3, mice) performed on three brain regions: stimulated focus with ($F$ = 5.19) and without FTT correction ($F$ = 5.51), out of focus ($F$ = 2.36, blue) and auditory region ($F$ = 0.68, orange). $P$ values indicated by labels. See Supplementary Fig. 7 for a more complete mouse brain atlas reference.

is the intensity at the spatial peak pulse averaged). The $Ca^{2+}$ movies were processed using a dedicated analysis pipeline[28] that compensates for anaesthesia-related slow resting-state activity[31] (Fig. 2a) and an evoked fluorescence-thermal dip[29], whose impact on the fluorescence signal can be precisely anticipated (Fig. 2b). The residual $Ca^{2+}$ signals show a sharp dose-dependent excitation region exactly at the point of TUS delivery (red spot in Fig. 2a,b) at the primary somatosensory cortex−lower limb region (anterior−posterior approximately −1.2 mm from bregma, medial−lateral approximately −1.6 mm).

We explored the activation thresholds by changing the peak pressure delivered in mouse cortex (Fig. 2c). The $Ca^{2+}$ responses reliably rise above the background at pressures exceeding 2.5 MPa, substantially higher than, for example, response thresholds observed recently for a much larger single-focus TUS at 0.6 MHz (ref. 20). Note that the sonication parameters used in the current study have previously been shown to cause no brain damage or other irreversible effects[29]. One-way analysis of variance (ANOVA) ($n$ = 3) indicates a strong dependency on the peak pressure of the (FTT-corrected) area under the curve at the point of US delivery (Fig. 2d, left). By contrast, no statistically significant dependency was found in the auditory and off-target motor regions (Fig. 2d, right). Based on the peak fluorescence data (Supplementary Fig. 8b), fluorescence in the auditory region does not significantly change with TUS pressure ($P$ = 0.758). In addition, off-target peak fluorescence remains consistently below 0.5% for pressures exceeding 2.3 MPa, in agreement with a shorter time to peak (Supplementary Fig. 8c) at the stimulated region.

**TUS steering and holographic enhancement of focal responses**
Electronic steering successfully moves the activation area, eliciting localized $Ca^{2+}$ activation responses (Fig. 3a). Although minor crosstalk up to 40% is observed in the time traces, the peak activation occurs at the targeted location with a pressure of 2.8 MPa. By stark contrast,

using hTUS for mouse cortex stimulation reveals a notable distributed excitation effect, achieving a robust activation response at only 1.2 MPa (Fig. 3b,c) with a triangular pattern where the foci centres are positioned at a 0.5 mm radius, enabling a clear separation between each focal spot (Fig. 1c).

Upon further exploration of hTUS responses, we found that pentagonal (green) and triangular (blue) hTUS patterns (0.5 mm radius) show clear activation responses already at a pressure of 0.9 MPa (Fig. 4a). At 1.1 MPa, pentagonal and triangular hTUS-triggered activation appear almost identical; nevertheless, the size of the pattern does have an effect: for example, the larger triangular hTUS pattern (purple, 1 mm radius) excites the mouse cortex at 1.2 MPa but with a lower activation amplitude than its 0.5 mm radius counterpart. Overall, all hTUS patterns tested induced a detectable and consistent $Ca^{2+}$ activation response at remarkably lower pressures compared to single-focused TUS. When compiling data from both the single-focus TUS and hTUS based on peak pressure, a nuanced scenario emerges (Fig. 4b) indicating distinct pressure activation thresholds for TUS and hTUS.

Examining the activated area as a function of the acoustic power (Fig. 4c) illustrates how the US field can lead to different outcomes for the same acoustic power. TUS triggers a $Ca^{2+}$ response only at powers exceeding 0.1 W, whereas triangular and pentagonal hTUS patterns of stimulation achieve an activated area of approximately 2 $mm^2$ at around 0.1 W (Methods, 'Fluorescence data processing' and 'Ultrasound excitation model'). Notable differences are observed between large (purple) and small (blue) triangular hTUS patterns at 70 mW. Statistical analysis shows no significant difference in the maximum activation areas between hTUS and TUS (one-way ANOVA, $F$ = 0.78, $P$ = 0.54, $n$ = 3).

**US excitation model beyond peak pressure**
Could TUS and hTUS activation differences be explained by more sophisticated US field descriptors? To address this question, we developed a

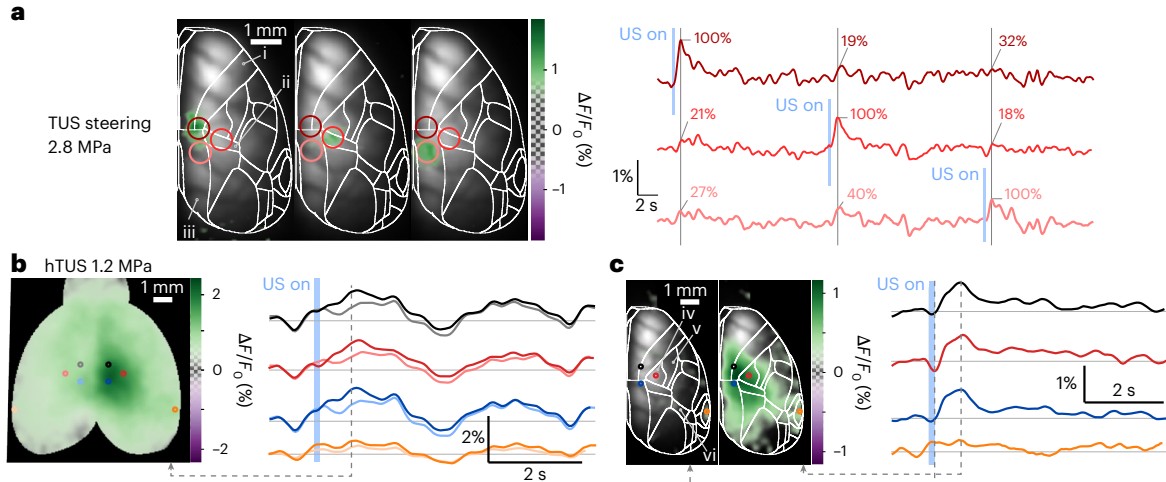

**Fig. 3 | TUS wavefront shaping enables dynamic probing of distributed networks. a**, Snapshot of Ca²⁺ changes in the mouse brain and time traces at selected points for TUS beam steering. The total stimulation period corresponds to 30 s (10 s between foci) repeated 10 times. Percentages on the time traces represent the relative fluorescence amplitude with respect to the point of stimulation (100%). Anatomical labels: i, primary motor area; ii, primary somatosensory area (trunk); iii, retrosplenial area. **b**, The corresponding changes for hTUS stimulation. Ultrasound pulse duration shown as light blue region. **c**, Resting-state-corrected snapshots. Traces from **a** and **b** are corrected for slow resting state activity. Anatomical labels: iv, primary somatosensory area (lower limb); v, primary somatosensory area (upper limb); vi, rostrolateral visual area.

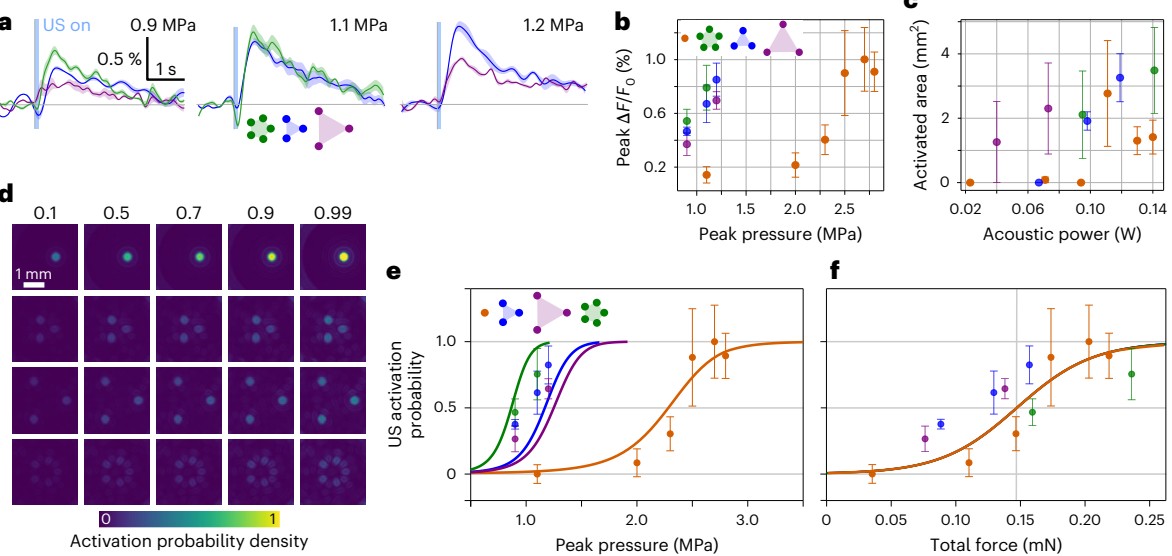

**Fig. 4 | hTUS effectively activates neural networks. a**, Progression of the Ca²⁺ signal with increasing peak pressure at the centre of the hTUS patterns (without FTT correction). US pulse duration is represented as a light blue region. Curves correspond to the mean ± s.e.m. **b**, Peak activation as a function of peak pressure. Data points correspond to the mean ± s.e.m. Colours correspond to the labels at the top of the panel. **c**, Activated area ($\Delta F/F_0 > 0.5\%$ around the centre of the sonicated point, 0.46 s < time < 1 s) between TUS (red) and hTUS as a function of acoustic power. Points represent the mean, while error bars correspond to the ±s.e.m. for each. **d**, APD calculated from the TUS (top row) and hTUS simulated fields for an activation probability indicated by the labels on top. **e,f**, Activation probability and experimental data (mean ± s.e.m) from **b** as a function of peak pressure (**e**) and total force (**f**). All experimental data obtained from $n$ = 3 mice by averaging 20 stimulations.

US excitation model (USEM; Methods and Supplementary Methods) that accounts for the physical properties of the hTUS delivery beyond the pressure–frequency dyad. We hypothesized that neural activation is influenced by specific US parameters, that is, radiation force or pressure, and established an activation probability density (APD) for both TUS and hTUS (Fig. 4d). Integrating the APD into a sigmoid function allows for predictions based on the peak pressure (Fig. 4e). Using a single set of parameters, USEM thus predicts distinct activation thresholds for TUS and hTUS configurations (Fig. 4e), showing qualitative agreement with hTUS observations. Interestingly, a linear relationship between radiation

force density and APD aligns well with USEM predictions (Fig. 4f), suggesting a unified threshold of ~0.15 mN, which matches the experimental data. Attempts to predict activation thresholds using other parameters like pressure or acoustic power and considering the negative influence of temperature resulted in a higher mean square error (Supplementary Figs. 4–6 and Supplementary Table 1).

## A physiological basis for the TUS and hTUS response

Could the differences in activation between TUS and hTUS stem from the connectivity of the underlying cortical population? We constructed

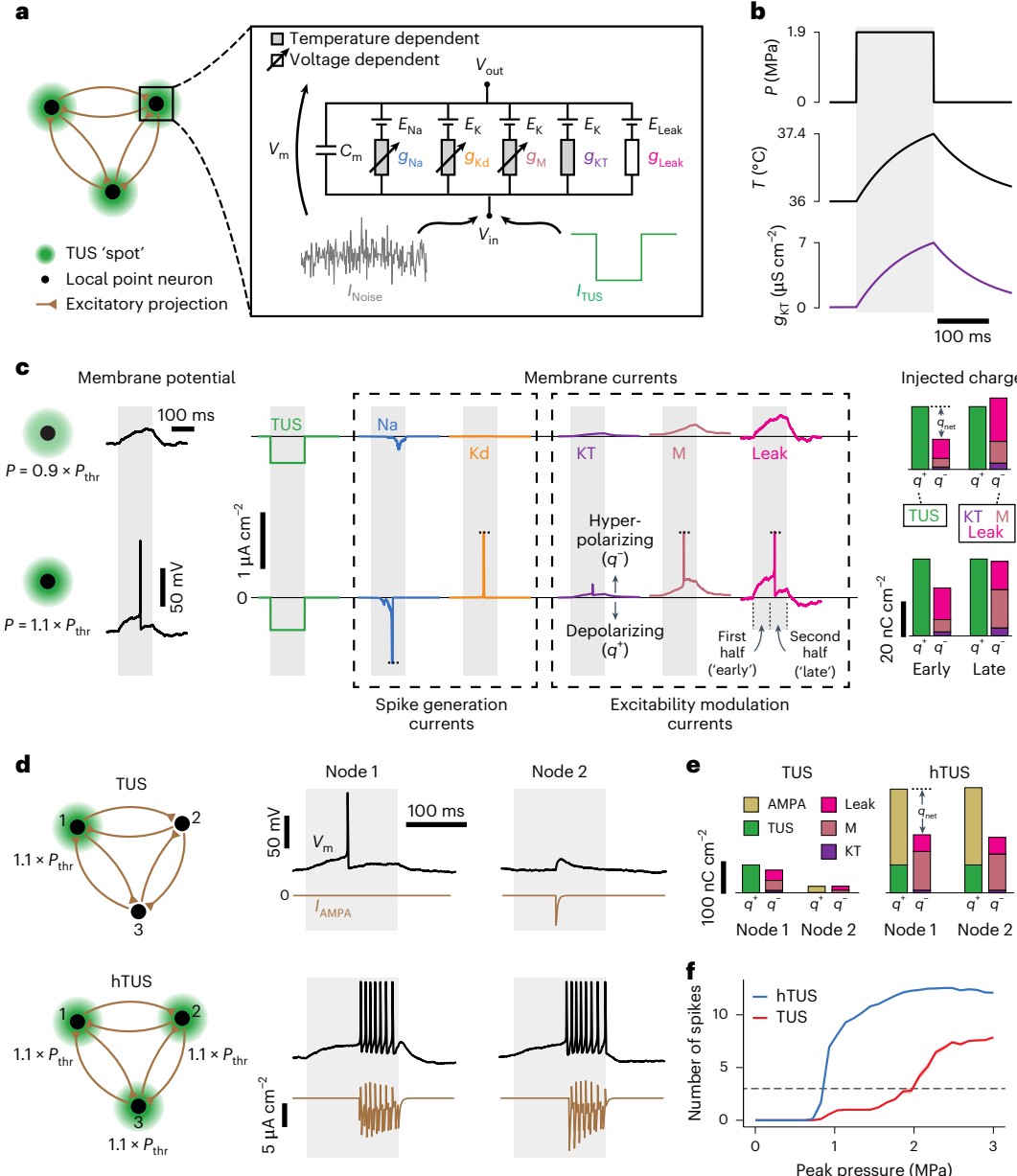

**Fig. 5 | A cortical network model predicts distinct response thresholds for TUS and hTUS. a**, Schematic description of the sCNM, illustrating the network architecture (left) and the point-neuron electrical circuit at each network node (right), where the transmembrane potential ($V_m = V_{in} − V_{out}$, where $V_{in}$ and $V_{out}$ refer to intracellular and extracellular voltages, respectively) across the membrane capacitor (capacitance $C_m$) is regulated by a set of ionic currents each associated with a specific conductance ($g$) and reversal potential ($E$). **b**, Time course of model internal variables upon application of a 150-ms-long 1.9 MPa TUS stimulus. From top to bottom: peak pressure amplitude ($P$), local temperature ($T$) and conductance of thermally activated potassium current ($g_{KT}$). **c**, Left: time course of local transmembrane voltage in an isolated network node induced by a 150-ms-long TUS stimulus at 0.9 (top) and 1.1 (bottom) times the threshold activation pressure ($P_{thr}$); fast membrane potential fluctuations were driven by a Gaussian-noise current with an amplitude of 2 μA cm$^{-2}$; the grey area marks the stimulation window. Middle: time course of transmembrane currents (bounded to ±1.2 μA cm$^{-2}$) for the same stimuli, including the TUS-induced depolarizing current ($I_{TUS}$), spike generation currents ($I_{Na}$ and $I_{Kd}$) and excitability modulation currents ($I_{KT}$, $I_M$ and $I_{Leak}$); by convention, negative currents are depolarizing, while positive currents are hyperpolarizing. Right: breakdown of the cumulative charge injected by depolarizing currents ($q^+$, that is, $I_{TUS}$) and hyperpolarizing currents ($q^−$, that is, $I_{KT}$, $I_M$ and $I_{Leak}$) during the first and

second halves of the stimulation window (referred to as 'early' and 'late' phases, respectively), illustrating the increased influence of excitability modulation currents as the stimulus progresses. Spike-generating currents were voluntarily omitted to specifically examine changes in nodal excitability. **d**, Time course of transmembrane voltages (in black) and synaptic currents ($I_{AMPA}$, in brown) in two representative nodes of a fully connected three-node network, upon application of a 150 ms stimulus at 1.1 times the threshold activation pressure, either to a single network node (top, conventional TUS) or to all three network nodes (bottom, hTUS). **e**, Breakdown of the cumulative charge injected by depolarizing currents ($q^+$, that is, $I_{TUS}$ and $I_{AMPA}$) and by hyperpolarizing currents ($q^−$, that is, $I_{KT}$, $I_M$ and $I_{Leak}$) during the stimulus window for the same representative nodes, illustrating the role of the excitatory synaptic current ($I_{AMPA}$) in maintaining a depolarized state enabling sustained firing in the hTUS case. **f**, Number of evoked spikes in sonicated nodes as a function of peak pressure amplitude, for both TUS (red) and hTUS (blue) 150 ms stimulation. Traces and shaded areas denote the mean ± s.e.m. of 50 simulations where membrane potential fluctuations were driven by a Gaussian-noise current. The dashed horizontal line denotes a theoretical fluorescence detection threshold. Currents nomenclature is as follows: Na, sodium; Kd, delayed rectifier potassium; M, slow non-inactivating potassium; KT, thermally activated potassium; AMPA, AMPA receptor post-synaptic current; Leak, non-specific leakage.

a simplified cortical network model (sCNM; Methods and Supplementary Table 2), where local neuronal populations within each TUS 'spot' are represented by single-point neuron models, which are interconnected via a set of excitatory projections (Fig. 5a). To account for TUS-evoked spiking activity, we incorporated a stimulus-dependent excitatory drive at each 'node'. In addition, we considered the thermal impact induced by TUS on membrane dynamics, acting via a temperature-dependent modulation of channel gating kinetics and conductances[32] as well as via a specific thermally activated inhibitory current[33] (Fig. 5b).

We first examined the model's behaviour on an isolated node, that is, in the absence of network contributions. In this configuration, the sonication onset triggers an immediate activation of the TUS-dependent current, leading to a gradual membrane depolarization (Fig. 5c, top). Above critical acoustic pressure, the node eventually reaches its spiking threshold and fires an action potential through the combined action of voltage-gated sodium and potassium currents (Fig. 5c, bottom). However, as the stimulus progresses, TUS-driven depolarization and spiking activity become increasingly countered by (1) non-specific leakage currents and (2) the activation of outward potassium currents due to rises in voltage and temperature. These combined inhibitory forces restrict membrane depolarization and therefore prevent subsequent spiking during the second half of the stimulus (Fig. 5c, bottom right).

We then examined the responses evoked by TUS and hTUS stimuli in a fully connected three-node cortical network model mimicking our empirical configuration. When a supra-threshold stimulus is applied to a single node (conventional TUS case), the induced spiking activity generates post-synaptic currents that are too weak to activate non-sonicated nodes (Fig. 5d, top). Conversely, when the same stimulus is applied to all three nodes (hTUS case), the concurrent activation of multiple interconnected nodes leads to a convergence of pre-synaptic inputs at each node. This results in larger post-synaptic currents (Fig. 5e) that amplify the initial neuronal response. Consequently, neurons enter a substantially higher spiking regime even at just-supra-threshold pressures (Fig. 5d, bottom). This sharp transition allows for clear response differentiation between TUS and hTUS across a wide acoustic pressure range, effectively reducing the detectable response threshold by more than half (Fig. 5f).

## Discussion

We report on neuromodulatory effects of precision-controlled and holographically distributed US delivery in the living mouse brain. Our method achieves precise focusing suitable for the dimensions of the mouse brain, closely approximating the relative focal dimensions used in targeting the human brain with TUS (Supplementary Table 3). Through precise stimulation coupled with concurrent cortex-wide Ca²⁺ imaging and a meticulous correction scheme, we have demonstrated robust localized Ca²⁺ responses to US stimuli. These responses are clearly distinguishable from auditory[21] and sensory confounds[34] or other indirect effects[35] and can be dynamically shifted by electronic steering. Our analysis further reveals distinct activation thresholds for TUS compared to hTUS, uncovering an excitability regime not fully explained by the conventional pressure (intensity)–frequency dyad[9,10].

To better understand these previously unexplored US neuromodulation aspects, we developed a dual modelling framework, where both an empirical and a mechanistic model were constructed to capture the intricacies of hTUS. These models achieve qualitative agreement with our experimental findings (Figs. 4e,f and 5c) while illuminating the complexity of the in vivo results. The modelling approach has limitations. First, the nature and strength of the direct effects induced by TUS on cellular membrane dynamics and the resulting neural activation are currently unclear[18,36–39]. Our sCNM model therefore assumes a simplistic mechanoelectrical transduction whereby TUS acoustic pressure is directly converted into a depolarizing membrane current.

This mechanism-agnostic strategy captures the pressure dependency of TUS-evoked responses through a single parameter, which can be tuned to replicating our empirical excitation thresholds. Second, the mechanisms by which TUS-evoked temperature variations affect neural activity remain to be elucidated. We therefore pursued various strategies to incorporate temperature sensitivity in our sCNM model based on established literature[32,40] as well as recent empirical evidence in mouse cortical and striatal neurons[33] (Methods). Simulations in our thermally sensitive model indicate that temperature variations induced in our experiments have minimal impact on neural activity and that voltage variations and spiking activity itself play a much more predominant role in regulating neural excitability (Fig. 5c). Finally, our sCNM model adopts a simple network architecture consisting of excitatory-only point-neuron models in a fully connected network. While this architecture does not represent the full complexity of cortical circuits, it nonetheless captures essential features of their organization that are relevant in the context of our study. For instance, there is substantial evidence that mid-range synaptic projections—such as those occurring between TUS hotspots—are primarily excitatory[41]. Hence, despite several simplifications, our sCNM provides valuable insights into the network mechanisms enabling enhanced cortical responses to hTUS.

Our findings contribute to the ongoing efforts towards understanding the biophysics of US neuromodulation. Previous studies have interpreted the frequency dependence of TUS as evidence for different mechanistic hypotheses, such as the predominance of radiation force at high frequencies[19] versus nano-cavitation at lower frequencies[36,42]. Our results underscore the importance of carefully decoupling the effects of the spatial extent of US stimulation from those related to frequency. We introduce the total radiation force as an empirical metric for assessing US interactions with the brain (Fig. 4f), indicating that radiation force distribution is a more pertinent factor than peak pressure. The USEM framework uses linear wave propagation simulations based on calibrated transcranial pressure measurements followed by a momentum flux density tensor[43] approach in an inviscid fluid to retrieve the radiation force. While efforts continue to refine these calculation methods[44], further experimental work is needed to validate our results and accurately measure the radiation force in brain tissue.

Although lower frequencies capable of penetrating the human skull have been applied to mice[20], this approach is not capable of generating tightly localized responses. Focal excitation at similar frequencies to ours (2 MHz) combined with cellular resolution recordings[45] has added a complementary view by showing the dependence of TUS on $I_{SPTA}$ (intensity at spatial peak temporal average). It remains to be seen how our observations translate to sub-megahertz range used to traverse the human skull.

Finally, our findings suggest that exploring holographic US fields to interrogate brain networks offers a promising alternative to the conventional single-focus approaches[46,47] in both therapy and basic neuroscience, thus opening avenues for the development of innovative US delivery strategies for both preclinical and human settings.

## Methods

### The integrated US stimulation fluorescence imaging set-up

US stimulation is delivered by 512 channels matrix array (Fig. 1a) distributed over a spherical surface encompassing 150° (Imasonic SaS). The same array is used in a fully reciprocal detection mode to find the stimulation spot in the mouse brain by means of volumetric optoacoustic tomography[29]. The excitation of optoacoustic responses is performed via three lateral apertures that provide illumination from a tunable pulsed optical-parametric-oscillator laser source (InnoLas Laser) generating nanosecond duration pulses at 800 nm wavelength and 10 mJ per-pulse energy. A custom-design multichannel electronic system (Falkenstein Mikrosysteme) drives and digitizes the US array signals with a 5.5 ns temporal resolution.

hTUS stimulation is achieved by splitting the 512 elements of the array into randomly distributed subsets. Each array's subset is assigned a different focal position forming a triangle (170 elements per focal point) or a pentagon (102 elements per focal point). To aim at different points in space, the time of flight between the selected focus and the array subset's elements is calculated and the relative delays between elements applied to the 3 MHz main waveform. A personal computer controls the hTUS emission by means of a graphical user interface implemented in MATLAB (2020, MathWorks).

Fluorescence imaging at 20 Hz is performed through a fibrescope (Zibra Corporation) mounted on the array's vertical aperture to record calcium dynamics from the mouse brain[48]. Excitation light for fluorescence imaging is generated by a continuous wave laser (Sapphire LPX, Coherent Europe) delivering 150 mW power at 488 nm. An optical filter (MF525-39, Thorlabs) rejects the excitation light with only the fluorescent light reaching the EMCCD camera (iXon Life, Andor). Image acquisition is performed via Andor Solis software (version 4.31).

US emission and fluorescence data acquisition are synchronized using a common trigger (DG5072, Rigol).

For optimal US coupling, the spherical array transducer is placed inside a small water tank filled with degassed and de-ionized water. The tank has a 6 cm diameter opening at its bottom sealed with a thin transparent plastic membrane to allow for an unobstructed passage of US and light waves transcranially into/from the mouse brain.

### Fluorescence data processing
Epi-fluorescence calcium recordings performed at 20 f.p.s. were Kalman-filtered (sigma = 0.5, Fiji[49] version 1.53c). A moving baseline, calculated as the 10th percentile in segments of 500 frames, is further applied to remove signal drifts due to laser energy fluctuations and photobleaching (MATLAB, 2020, MathWorks).

Further postprocessing is performed using Python (Python Software Foundation, Python Language Reference, version 3.11.6; available at http://www.python.org). Twenty consecutive stimulations were averaged to generate a 10 s stack. To reduce the spatial noise, a Gaussian filter with standard deviation of 3 pixels (~120 μm) was applied to the cycle-averaged image stack.

Extracted time traces from 1 pixel are temporally smoothed using second-order Savitzky–Golay filter of 11 samples in length. The traces from different mice are added to obtain mean traces and their respective standard error of the mean (Figs. 2c and 4a).

The activated area is calculated using a flood-fill algorithm starting at the focal point for TUS or at the centre of the hTUS pattern. The area is then calculated as the pixels fulfilling relative fluorescence change ($\Delta F/F_0$) > 0.5% at any time between 0.46 s and 1 s after the start of the US stimulus.

### FTT correction
The bioheat equation as described by Pennes[50] is solved using a previously reported implementation[51] with the focal US intensity calculated using Field II (version 3.30, http://field-ii.dk/) as the heat source. As the thermal modelling does not yield absolute temperature predictions but rather a relative spatio-temporal evolution of it, we scaled the temperature transient at the focus centre using the $\Delta F/F_0$ values at $t = 0$ and $t = 150$ ms, inverted its sign to match the fluorescence quenching and subtracted it from the $Ca^{2+}$ time trace. Temporal smoothing was applied after FTT correction[28].

### hTUS characterization
The focusing capabilities of the spherical array are characterized in a water tank using a 75 μm diameter calibrated hydrophone (Precision Acoustics). A two-dimensional scan of the hydrophone (Fig. 1c) portrays the pressure distribution, in good agreement with the prediction of a linear model calculated using Field II. Characterization of the TUS and hTUS pressure field is shown in Supplementary Fig. 3.

Measurement of the transcranial pressure using an excised mouse skull was performed for different voltages and frequencies (Fig. 1e and ref. 29) and was used to calibrate the ultrasound simulation.

The FTT, described in detail elsewhere[29], can be used to characterize the focusing of hTUS in vivo through an intact mouse skull. Following the US stimulation, localized mild temperature rise quenches the GCaMP6f fluorescence resulting in a negative signal change. Delivering the US simultaneously to three different points (170 elements per focus) generates a mild negative change (Fig. 1c). However, steering all 512 elements sequentially and using minimum intensity projection over time renders a more substantial change. Both examples show that the shape of the field is still retained after transcranial propagation.

### Animal experiments
All procedures involving mice conformed to the national guidelines of the Swiss Federal Act on animal protection and were approved by the Cantonal Veterinary Office Zurich. Animals were housed in individually ventilated cages inside a temperature-controlled room, under a 12 h dark/light cycle. Pelleted food (3437PXL15, CARGILL) and water were provided ad libitum.

A total of 10 GCaMP6f mice (C57BL/6J-Tg(Thy1-GCaMP6f) GP5.17Dkim/J, The Jackson Laboratory) were used for this study (4 female, 6 male). The mouse head was immobilized using a custom stereotactic frame (Narishige International). Blood oxygen saturation, heart rate and mouse body temperature were continuously monitored (PhysioSuite, Kent Scientific), and the body temperature was kept within physiological range with a heating pad. The hair on the mouse head was first trimmed and then removed using shaving cream to ensure optimal US coupling after scalp removal. The mice were subcutaneously injected with 0.1 mg kg$^{-1}$ buprenorphin 30 min before scalp removal. A 40% dilution of phosphate-buffered saline in US gel (Aquasonic clear, Parker Laboratories) was deposited on the mouse head and brought into contact with the transparent membrane of the water tank to ensure unobstructed delivery of light and ultrasound into the mouse brain.

The mice were sonicated under isoflurane anaesthesia (3% $v/v$ for induction, 1.2% $v/v$ for maintenance) with 150 ms duration pulses at 3 MHz delivered sequentially every 10 s a total of 20 times.

### Ultrasound excitation model
This USEM is based on the ad hoc fitting of sigmoid function, whose parameters depend on the ultrasound pressure field (Supplementary Figs. 3–6). We first use linear acoustic simulations of the array based on the hydrophone measurements through a mouse skull using a spatial grid size of $40 \times 40 \times 40$ μm$^3$ and temporal step of 0.05 μs. The high spatio-temporal sampling allows the calculation of the acoustic velocity and intensity. The radiation force density is calculated using the full vectorial expression found in equation (12) of ref. 43, which is later integrated along all three spatial dimensions to yield the total radiation force (Supplementary Equation (20)). Similarly, the acoustic power is obtained by integrating the axial acoustic intensity ($I_z$) along the $xy$ plane at the focus. In addition, excitation distribution is calculated by evaluating excitability functions that depend on the pressure, temperature (from the intensity) and radiation force at any given point in space (Supplementary Methods). The function evaluated at the focal plane is integrated over $3 \times 3$ mm$^2$ area and the outcome used as a variable of a sigmoid function to yield the activation probability (see Supplementary Methods for more details).

### sCNM
We modelled the electrophysiological behaviour of local cortical populations using representative conductance-based point-neuron models describing the temporal evolution of the local membrane potential $V_m$ at each node.

In the absence of stimulus, local membrane dynamics were captured by the canonical set of transmembrane ionic currents found in cortical regular spiking neurons[52], that is, a sodium current ($I_{Na}$), delayed-rectifier and slow non-inactivating potassium currents ($I_{Kd}$ and $I_M$, respectively), and a non-specific leakage current ($I_{Leak}$), each described by specific Hodgkin–Huxley formalisms[52]. Stochastic fluctuations in membrane potential were incorporated by adding a zero-mean Gaussian noise current ($I_{Noise}$) to desynchronize neural activity across nodes. Excitatory connections between each node pair were modelled with an AMPA (α-amino-3-hydroxy-5-methyl-4-isoxazolepropionic acid)-like synaptic current ($I_{AMPA}$), turned on by the pre-synaptic node crossing a voltage threshold[53]. The synaptic weight of node-to-node connections was tuned to obtain gradual network entrainment upon emergence of robust spiking activity. The stimulus-dependent excitatory drive $I_{TUS}$ was defined as a constant depolarizing current proportional to the peak pressure amplitude, with a scaling factor tuned to replicate the single-focus empirical response threshold (around 2.2 MPa), considering that a minimum of $n = 3$ spikes would be necessary to generate a detectable change in GCaMP6f fluorescence.

Local TUS-induced temperature variation was modelled using a first-order exponential convergence scheme, with a fixed time constant and a steady-state value depending linearly on the instantaneous TUS intensity, namely, $I_{SPPA} = P^2/(2Z)$, where $P$ is the pressure amplitude and $Z$ is the medium's characteristic acoustic impedance. Thermal parameters were tuned to best replicate predictions of the full thermal model and experimental data.

Temperature sensitivity was incorporated into various features of the model. First, the gating kinetics of voltage-gated channels was modulated by a temperature-dependent factor, using a $Q_{10}$ formalism as in previous studies[32,40]. Second, the maximal conductance of sodium and potassium channels followed an analogous $Q_{10}$-based modulation, with coefficients fitted to experimental data[32]. Third, following recent experimental evidence[33], a specific outward potassium current ($I_{KT}$ with a conductance proportional to temperature increase was added to the model. All model parameters are given in Supplementary Table 2.

The model was implemented using NEURON version 8.2.2 (https://www.neuron.yale.edu/neuron/) and its Python interface.

### Reporting summary

Further information on research design is available in the Nature Portfolio Reporting Summary linked to this article.

## Data availability

Fluorescence data are available via ETH Zurich at https://doi.org/10.3929/ethz-b-000731630 (ref. 54). Source data are provided with this paper.

## Code availability

Code is available for fluorescence data processing via Zenodo at https://doi.org/10.5281/zenodo.15225873 (ref. 55) and for sCNM simulations at https://doi.org/10.5281/zenodo.15232784 (ref. 56).

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

## Acknowledgements

We thank Z. Chen and S. Shaykevich for helpful discussions and T. Warner for proofreading the paper. We also acknowledge the valuable help of M. Reiss with the mouse experiments. We acknowledge the following funding: National Institutes of Health grants RF1-NS126102 (S.S. and D.R.) and R01-NS109885 (S.S.), Swiss National Science Foundation grant 310030_192757 (D.R.) and Swiss National Science Foundation Postdoc-Mobility grant P500PB_211119 (T.L.).

## Author contributions

D.R. and S.S. conceived the experimental system. H.E. and A.Ö. developed the experimental system. H.E., N.D., Q.P. and Y.C. carried out experiments in mice with the help of A.Ö. H.E. developed the software for data analysis. H.E., Y.C., Q.P. and N.D. performed data analysis and visualization. S.S. and T.L. conceived the cortical network model, and T.L. developed the model, performed simulations and wrote the corresponding paper sections. H.E., D.R. and S.S. supervised the study. H.E. wrote the first draft of the paper. All authors reviewed and edited the paper.

## Funding

## Competing interests

The authors declare no competing interests.

## Additional information

**Correspondence and requests for materials** should be addressed to Shy Shoham or Daniel Razansky.

[1]Institute of Pharmacology and Toxicology and Institute for Biomedical Engineering, Faculty of Medicine, University of Zurich, Zurich, Switzerland. [2]Institute for Biomedical Engineering, Department of Information Technology and Electrical Engineering, ETH Zurich, Zurich, Switzerland. [3]Neuroscience and Ophthalmology Departments, NYU Langone Health, New York, NY, USA. [4]Tech4Health Institute, NYU Langone Health, New York, NY, USA. [5]Department of Neurosurgery, Lucerne Cantonal Hospital, Lucerne, Switzerland. [6]Department of Biomedical Engineering, New York University, New York, NY, USA. [7]Present address: School of Physics Science and Engineering, Tongji University, Shanghai, China. ✉e-mail: shoham@nyu.edu; daniel.razansky@uzh.ch

# Reporting Summary

## Statistics

For all statistical analyses, confirm that the following items are present in the figure legend, table legend, main text, or Methods section.

| n/a | Confirmed | |
|---|---|---|
| ☐ | ☒ | The exact sample size (*n*) for each experimental group/condition, given as a discrete number and unit of measurement |
| ☐ | ☒ | A statement on whether measurements were taken from distinct samples or whether the same sample was measured repeatedly |
| ☐ | ☒ | The statistical test(s) used AND whether they are one- or two-sided<br>*Only common tests should be described solely by name; describe more complex techniques in the Methods section.* |
| ☒ | ☐ | A description of all covariates tested |
| ☒ | ☐ | A description of any assumptions or corrections, such as tests of normality and adjustment for multiple comparisons |
| ☐ | ☒ | A full description of the statistical parameters including central tendency (e.g. means) or other basic estimates (e.g. regression coefficient) AND variation (e.g. standard deviation) or associated estimates of uncertainty (e.g. confidence intervals) |
| ☐ | ☒ | For null hypothesis testing, the test statistic (e.g. *F*, *t*, *r*) with confidence intervals, effect sizes, degrees of freedom and *P* value noted<br>*Give P values as exact values whenever suitable.* |
| ☒ | ☐ | For Bayesian analysis, information on the choice of priors and Markov chain Monte Carlo settings |
| ☒ | ☐ | For hierarchical and complex designs, identification of the appropriate level for tests and full reporting of outcomes |
| ☒ | ☐ | Estimates of effect sizes (e.g. Cohen's *d*, Pearson's *r*), indicating how they were calculated |

*Our web collection on statistics for biologists contains articles on many of the points above.*

## Software and code

Policy information about availability of computer code

| Data collection | Custom Matlab 2020 code for data collection, Andor Solis software (v4.31). |
|---|---|
| Data analysis | Custom Matlab 2020 and Python (v3.11.6) code for data analysis. 10.5281/zenodo.15225873. Fiji (v1.53c). NEURON (v8.2.2) |

For manuscripts utilizing custom algorithms or software that are central to the research but not yet described in published literature, software must be made available to editors and reviewers. We strongly encourage code deposition in a community repository (e.g. GitHub). See the Nature Portfolio guidelines for submitting code & software for further information.

## Data

Policy information about availability of data

All manuscripts must include a data availability statement. This statement should provide the following information, where applicable:
- Accession codes, unique identifiers, or web links for publicly available datasets
- A description of any restrictions on data availability
- For clinical datasets or third party data, please ensure that the statement adheres to our policy

10.3929/ethz-b-000731630

# Research involving human participants, their data, or biological material

Policy information about studies with [human participants or human data](). See also policy information about [sex, gender (identity/presentation), and sexual orientation]() and [race, ethnicity and racism]().

| | |
|---|---|
| Reporting on sex and gender | N/A |
| Reporting on race, ethnicity, or other socially relevant groupings | N/A |
| Population characteristics | N/A |
| Recruitment | N/A |
| Ethics oversight | N/A |

Note that full information on the approval of the study protocol must also be provided in the manuscript.

# Field-specific reporting

Please select the one below that is the best fit for your research. If you are not sure, read the appropriate sections before making your selection.

☒ Life sciences  ☐ Behavioural & social sciences  ☐ Ecological, evolutionary & environmental sciences

For a reference copy of the document with all sections, see [nature.com/documents/nr-reporting-summary-flat.pdf](nature.com/documents/nr-reporting-summary-flat.pdf)

# Life sciences study design

All studies must disclose on these points even when the disclosure is negative.

| | |
|---|---|
| Sample size | Exploratory study. Experiments where repeated until a sample size of n=3 was reached for every group. |
| Data exclusions | No data was excluded from the analysis. |
| Replication | Experiments where repeated until a sample size of n=3 was reached for every group. |
| Randomization | We consider the stimulation events independent due to its short duration and period between stimuli. No randomization was performed. |
| Blinding | Blinding is not relevant in our study. Changes introduced in the variable are controlled without altering the measurement setup. |

# Reporting for specific materials, systems and methods

We require information from authors about some types of materials, experimental systems and methods used in many studies. Here, indicate whether each material, system or method listed is relevant to your study. If you are not sure if a list item applies to your research, read the appropriate section before selecting a response.

## Materials & experimental systems

| n/a | Involved in the study |
|---|---|
| ☒ | ☐ Antibodies |
| ☒ | ☐ Eukaryotic cell lines |
| ☒ | ☐ Palaeontology and archaeology |
| ☐ | ☒ Animals and other organisms |
| ☒ | ☐ Clinical data |
| ☒ | ☐ Dual use research of concern |
| ☒ | ☐ Plants |

## Methods

| n/a | Involved in the study |
|---|---|
| ☒ | ☐ ChIP-seq |
| ☒ | ☐ Flow cytometry |
| ☒ | ☐ MRI-based neuroimaging |

## Animals and other research organisms

Policy information about studies involving animals; ARRIVE guidelines recommended for reporting animal research, and Sex and Gender in Research

| | |
|---|---|
| Laboratory animals | 10 GCaMP6f mice were employed for this study (4 female, 6 male, 5 - 7 weeks old) |
| Wild animals | No wild animals were used in this study |
| Reporting on sex | We used female and male mice in our study. |
| Field-collected samples | No field collected samples are required for this study |
| Ethics oversight | All procedures involving mice conformed to the national guidelines of the Swiss Federal Act on animal protection and were approved by the Cantonal Veterinary Office Zurich. |

Note that full information on the approval of the study protocol must also be provided in the manuscript.

## Plants

| | |
|---|---|
| Seed stocks | N/A |
| Novel plant genotypes | N/A |
| Authentication | N/A |

