## [Peer Review File · Nature Biomedical Engineering]

Holographic transcranial ultrasound neuromodulation enhances stimulation efficacy by cooperatively recruiting distributed brain circuits

Corresponding Author: Professor Daniel Razansky

Version 0:

Decision Letter:

Dear Daniel,

Thank you again for submitting to *Nature Biomedical Engineering* your manuscript, "Holographic ultrasound enhances neuromodulation by engaging distributed brain circuits". The manuscript has been seen by three experts (and you will find two reports at the end of this message); unfortunately, one of the reviewers, who we chased for many weeks, failed to provide a report. I should also apologize for the much delayed relay of these reports (in fact, I thought I had forwarded them to you while I was waiting for the feedback from Reviewer #2).

You will see that the reviewers appreciate aspects the work, and that Reviewer #3 is unconvinced about the extent of the mechanistic findings and their utility, particularly from a human-brain-physiology standpoint. The reviewers also express concerns about the degree of support for the claims. We hope that with substantial further work you can address the criticisms and convince the reviewers of the merits of the study. In particular, we would expect that a revised version of the manuscript provides extended analyses and mechanistic insight, as well as a thorough reporting of the methodology.

When you are ready to resubmit your manuscript, please upload the revised files, a point-by-point rebuttal to the comments from all reviewers, the [reporting summary](https://www.nature.com/authors/policies/ReportingSummary.pdf), and a cover letter that explains the main improvements included in the revision and responds to any points highlighted in this decision.

Please follow the following recommendations:

- * Clearly highlight any amendments to the text and figures to help the reviewers and editors find and understand the changes (yet keep in mind that excessive marking can hinder readability).
- * If you and your co-authors disagree with a criticism, provide the arguments to the reviewer (optionally, indicate the relevant points in the cover letter).
- * If a criticism or suggestion is not addressed, please indicate so in the rebuttal to the reviewer comments and explain the reason(s).
- * Consider including responses to any criticisms raised by more than one reviewer at the beginning of the rebuttal, in a section addressed to all reviewers.
- * The rebuttal should include the reviewer comments in point-by-point format (please note that we provide all reviewers will the reports as they appear at the end of this message).
- * Provide the rebuttal to the reviewer comments and the cover letter as separate files.

We expect that you will be able to resubmit the manuscript within 20 weeks of receiving this message. If this is the case, you will be protected against potential scooping. Otherwise, we will be happy to consider a revised manuscript as long as the significance of the work is not compromised by work published elsewhere or accepted for publication at *Nature Biomedical Engineering*.

We hope that you will find the referee reports helpful when revising the work. Please do not hesitate to contact me should you have any questions.

Best wishes,

Pep

Pep Pàmies

Chief Editor, Nature Biomedical Engineering

Reviewer #1 (Report for the authors (Required)):

In this paper, the authors studied the application of holographic ultrasound in neuromodulation, and proved that transcranial ultrasound can achieve high-precision and dynamically controllable nerve stimulation, and holographic transcranial ultrasound stimulation (hTUS) can significantly reduce the neural activation threshold. The experimental results show that hTUS can effectively modulate local and mid-range neural network projection, revealing the potential of complex interaction mechanism between ultrasound and neural tissue. The results of the paper are of great significance to the progress of non-invasive brain network regulation technology, and provide new methods and perspectives for future neuroscience research and treatment of neurological diseases. However, there are still some concerns need to be solved.

Major concerns:

- 1) In the fluorescence image on the left of Figure 2B, why is the fluorescence intensity strong in the lower left corner? And after a period of ultrasonic stimulation, the intensity went away again?
- 2) In Figure 2C, the authors used sound pressures of 2.5 MPa or higher to show a more pronounced rise in Ca²⁺ response, significantly higher than the response threshold recently observed at 0.6 MHz for larger single-focus TUS. Can this phenomenon be understood as caused by the size of the ultrasound stimulation area, and the authors should comment on this statement? In addition, how can the authors prove that such high sound pressure will not cause damage to brain tissue or nerve cells?
- 3) In Figures 3B and 3C, The authors mention that "In stark contrast, using hTUS for mouse cortex stimulation reveals a significant distributed excitation effect, achieving a robust activation response at only 1.2 MPa (Fig. 3B, C) with a triangular pattern where the foci centers are positioned at a 0.5 mm radius". Why did the authors set it to 0.5 mm here?
- 4) In Page 4, the authors mentioned that "At 1.1 MPa, pentagonal and triangular hTUS-triggered activation appear almost identical, nevertheless, pattern geometry does have an effect: for example, the larger triangular hTUS pattern (purple, 1 mm radius) excites the mouse cortex at 1.2 MPa but with a lower activation amplitude than its 0.5 mm radius counterpart.". From the results, it seems to have less to do with the pattern geometry (since pentagon and triangle have almost the same activation effect) and more to do with the distance between US firing points, right?
- 5) In Page 4, the authors state that "TUS triggers Ca²⁺ responses only at powers above 0.1 W, whereas triangular and pentagonal hTUS stimulation modes achieve an activation area of about 2 mm² at around 0.1 W.". How was 0.1 W determined here and how was the 2 mm² calculated?
- 6) In Figure 4C, why is the $\Delta F/F_0$ set to greater than 0.5%?
- 7) How is the Total force of Figure 4F calculated?
- 8) In the discussion, the authors claim that "Our method achieves precise focusing suitable for the dimensions of the mouse brain, closely approximating the relative focal dimensions used in targeting the human brain with TUS. Why do the authors say it is very close to the relative focus size used for TUS to target the human brain?"
- 9) The focal spot of ultrasonic stimulation used in the system is 250 μm , which is smaller than that used in most studies. Is the smaller the focal spot of ultrasonic stimulation the better?
- 10) In this paper, the system uses a 3 MHz probe. Because the human skull is relatively thick, in order to improve the penetration depth and reduce the impact of scattering and reflection, the ultrasonic frequency is usually used between 200 kHz and 1.5 MHz. Why 3 MHz ultrasound used here? In addition, does the frequency of ultrasound affect the effect of regulation?

Minor concerns:

- 1) In the Figure caption of Figure 1, "... the a 512-element spherically focused array".
- 2) In the discussion, the "result" in the sentence "While efforts are being made to continue perfecting these computational methods, further experimental work is needed to validate our results and accurately measure radiation forces in the brain" should be "results".

3) In the supplementary document, Page 2, "In doing that, the number of elements used to generate a focus change from 512 for single focus to 512/3 ~ 170 for a triangle and 512/5 ~ 102." should add "for a pentagon" at the end of the sentence.

4) In the supplementary document, Page 2, The "LISTNUMLISTNUM" follows formula (1) should be deleted.

Reviewer #3 (Report for the authors (Required)):

The study by Estrada et al. describes the application of a holographic transcranial ultrasound stimulation (hTUS) technique to improve neuromodulation through engaging multiple brain circuits in anesthetized mice. The authors delivered 3MHz US using a 512 channels spherical matrix array to focus to ~250um X 500um, and validated focal activation by measuring cortical Ca responses using GCamp6f. A key observation is that sonicating multiple foci lowered the threshold of Ca activation. Using a USEM model, the study identified a unified Ca activation threshold of 0.15mN. Furthermore, the study attempted to evaluate network versus cellular effects using a cortical neural network model containing individual neurons modelled biophysically, and demonstrated thermal and voltage effects. Overall, this study presents a good solution to shape US focus to meet the challenge of the small head in mice. However, the impact of the work, especially, the translational impact is limited. The observation that sonicating multiple area, collectively a largely area, reduces activation threshold is interesting. The modeling attempt however cannot offer concrete mechanistic insights.

Major Comments:

1. It is unclear about the significance of the work. The introduction lacks detailed discussion on the existing gaps to justify the need for the current research. Although this tool enables better focusing for mice, 3 MHz US does not pass through the human cranium as well as sub-MHz frequencies. Could the authors comment more on how mechanistic results derived from these methods may differ from the lower frequencies typically used in human studies? For example, radiation force is more likely a mechanism at higher frequencies given its relationship with absorption.
2. The study claims precise focusing and steering capabilities but does not thoroughly discuss the limitations, particularly steering accuracy. Although the focal size (~250 μm laterally, 500 μm axially) is mentioned, it is not compared to focal sizes in other TUS studies in mice. Additionally, while the study states that the focal size to brain region ratio is similar to human studies, it lacks a detailed quantitative comparison between human and mouse brain targeting ratios.
3. Lacks control data that could serve as a baseline for comparison with the TUS-treated groups.
4. The study measures responses in the auditory region to identify potential auditory confounds. However, without detailed methodology, control experiments, and quantitative analysis, it is difficult to conclusively differentiate between direct neural activation and indirect auditory effects.
5. The study uses the area under the curve (AUC) to analyze Ca²⁺ responses in different brain regions following TUS, comparing the target site to auditory and off-target regions. While AUC measures overall activity, it can obscure temporal dynamics and peak response characteristics. A more detailed analysis of peak amplitude, response latency, and temporal patterns would help differentiate direct TUS effects from confounds.
6. Figure 3 mentions minor crosstalk observed in the time traces but does not provide a quantitative analysis of this crosstalk.
7. The assumptions made in developing the cortical network model and their potential limitations are not fully discussed.
8. While the network model in Figure 5 is a promising potential physiological mechanism, it is difficult to interpret the experimental differences between TUS and hTUS without addressing sonication area as a potential confounding factor. Could these differences be explained by a higher volume of sonicated tissue in the hTUS condition? It would be interesting to compare hTUS to a larger single focal spot TUS with similar focal volume to the hTUS condition (potentially by patterning multiple overlapping hTUS spots). This may help decouple the impact of sonicated tissue volume vs local network projections in influencing observed changes in activation threshold.
9. Why did the study use TUS parameters that induce tissue heating (i.e., continuous wave US)? How might thermal effects be decoupled from ultrasound-induced mechanical responses in this study?
10. It would be helpful to discuss and compare TUS and hTUS axial focal zones as well, as a key strength of this work is achieving precise TUS focality.
11. Why might hTUS be less precisely focused than steered TUS? Is it because hTUS creates more tissue heating?

Minor Comments:

1. The title is generally accurate but could benefit from slight adjustments to better reflect the specific enhancements demonstrated, the extent of mechanistic understanding, and the model used.
2. Duty cycle, pulse repetition frequency, pulse duration, and ISPPA/ISPTA are not listed.

3. Within figure 1, the ultrasound waveform used should be added.
4. Figure 1A lacks detailed labels and annotations (red / blue colors) that explain the key components.
5. The in vivo characterization shown in Figure 1C should specify the exact brain region, or at least within the figure caption.
6. The study measures Ca²⁺ signals in various regions of the mouse brain, including the target stimulation site and potential auditory regions, to assess neural activation induced by TUS (Figure 2). However, the specific regions of interest for these measurements are not clearly defined, dorso / lateral / etc, making it challenging to interpret the spatial specificity and relevance of the observed responses.
7. The time course of local transmembrane currents and membrane potentials (Figure 5B) is presented, but the traces are densely packed and difficult to read.
8. The methods lack detailed information on the calibration and validation of the ultrasound array to ensure accurate and consistent pressure fields.
9. The discussion sometimes overgeneralizes the results without adequately considering the limitations of the study.

Version 1:

Decision Letter:

Dear Professor Razansky,

Thank you for your revised manuscript, "Holographic transcranial ultrasound neuromodulation enhances stimulation efficacy by cooperatively recruiting distributed brain circuits". Having consulted with the original reviewers I am pleased to write that we shall be happy to publish the manuscript in *Nature Biomedical Engineering*.

We will be performing detailed checks on your manuscript, and in due course will send you a checklist detailing our editorial and formatting requirements. You will need to follow these instructions before you upload the final manuscript files.

Best wishes,

Barbara Cheifet
Interim Chief Editor
Nature Biomedical Engineering

Reviewer #1 (Report for the authors (Required)):

The authors have essentially addressed the issues that I raised in the revised manuscript. However, there are still a few minor issues that require further clarification.

1. Figure 1A clearly illustrates the optoacoustic setup, but it appears that no results related to optoacoustic imaging are presented in the manuscript. Although the authors refer to previous studies demonstrating the use of optoacoustic tomography for precise localization of the stimulation spot in the mouse brain, the absence of any direct optoacoustic results in this study raises questions about the relevance of this setup to the current work. I would recommend either: a) Including optoacoustic imaging results to demonstrate how this technique was used to localize the stimulation spot, thereby enhancing the completeness of the study; or b) Removing the optoacoustic setup from Figure 1 if it was not utilized in the current experiments, to avoid confusion and maintain focus on the primary methods and results of this study.

This clarification would help to streamline the presentation of the manuscript and ensure that all components of the experimental setup are relevant to the findings reported.

2. The authors stated, "Although minor crosstalk up to 40% is observed in the time traces, the peak activation occurs at the targeted location with a pressure of 2.8 MPa." However, is it appropriate to characterize 40% crosstalk as "minor" given that 40% is quite a significant value? It is recommended to remove the word "minor."

Reviewer #3 (Report for the authors (Required)):

The authors have addressed all my concerns.

Version 2:

Decision Letter:

Dear Professor Razansky,

I am happy to inform you that your manuscript, "Holographic transcranial ultrasound neuromodulation enhances stimulation efficacy by cooperatively recruiting distributed brain circuits", has now been accepted for publication in *Nature Biomedical Engineering*.

Over the next few weeks, the figures will be checked for production quality, the text edited to ensure that it conforms to house style, and the manuscript typeset.

Our Articles are published about 40 days after the acceptance date (we recommend that you inform your institutional press office of this timeframe), and you will be notified of the actual publication date a few days in advance. Articles can be published any working day of the week, and are pushed live shortly after 10 am London time.

Publishing agreement. You will be asked to digitally sign a publishing agreement (grant of rights). After the signed publishing agreement has been received, the proofs of the article will be sent to you for review. If you have any queries during the production process, or you cannot meet the requested deadline for returning the proofs, please contact rjsproduction@springernature.com.

Nature Biomedical Engineering is a Transformative Journal. Authors may publish their research with us through the traditional subscription access route, or make their paper immediately open access through payment of an article-processing charge. More [information about publication options](https://www.springernature.com/gp/open-research/transformative-journals) is available.

You may need to take specific actions to [comply](https://www.springernature.com/gp/open-research/funding/policy-compliance-faqs) with funder and institutional open-access mandates. If the work described in the accepted manuscript is supported by a funder that requires immediate open access (as outlined, for example, by [Plan S](https://www.springernature.com/gp/open-research/plan-s-compliance)) and your manuscript was originally submitted on or after January 1st 2021, then you should select the gold OA route. Authors selecting subscription publication will need to accept our standard licensing terms (including our [self-archiving policies](https://www.springernature.com/gp/open-research/policies/journal-policies)), and these will supersede any other terms that the author or any third party may assert apply to any version of the manuscript.

Acceptance of your manuscript is conditional on agreement, by all authors, with both our [media embargo](http://www.nature.com/authors/policies/embargo.html) and [confidentiality and pre-publicity](http://www.nature.com/authors/policies/confidentiality.html) policies. In particular, you may arrange your own publicity of the Article (for instance, through your institutional press office), as long as you ensure that journalists strictly adhere to the media embargo.

To assist you in disseminating the work, as soon as the Article is published you will be able to take advantage of the Springer Nature [SharedIt](https://www.springernature.com/gp/researchers/sharedit) initiative to [generate a unique shareable link to the Article](http://authors.springernature.com/share) that will allow anyone (with or without a subscription) to read it. Recipients of the link who are subscribers will also be able to download and print the PDF.

Thank you for having submitted this work to *Nature Biomedical Engineering*.

Best wishes,

Barbara Cheifet
Editor
Nature Biomedical Engineering

Point to point response to the Reviewer's comments

Reviewer #1 (Report for the authors (Required)):

In this paper, the authors studied the application of holographic ultrasound in neuromodulation, and proved that transcranial ultrasound can achieve high-precision and dynamically controllable nerve stimulation, and holographic transcranial ultrasound stimulation (hTUS) can significantly reduce the neural activation threshold. The experimental results show that hTUS can effectively modulate local and mid-range neural network projection, revealing the potential of complex interaction mechanism between ultrasound and neural tissue. The results of the paper are of great significance to the progress of non-invasive brain network regulation technology, and provide new methods and perspectives for future neuroscience research and treatment of neurological diseases. However, there are still some concerns need to be solved.

Reply: We thank the Reviewer for recognizing the significance and novelty of our work and hope they will find the revised version satisfactory.

Major concerns:

1) *In the fluorescence image on the left of Figure 2B, why is the fluorescence intensity strong in the lower left corner? And after a period of ultrasonic stimulation, the intensity went away again?*

Reply: We thank the Reviewer for pointing this out. We checked the data (Fig. below) and it corresponds to a single event at the second stimulation where the resting state becomes asymmetric (after correction) right before the stimulation takes place (bottom left, green trace). After excluding this outlier stimulation event from the analysis, the cycle average becomes much cleaner, which is now included in the revised Fig. 2B. When comparing the time traces (bottom right), only small changes can be observed between outlier-corrected and uncorrected cases.

2) In Figure 2C, the authors used sound pressures of 2.5 MPa or higher to show a more pronounced rise in Ca²⁺ response, significantly higher than the response threshold recently observed at 0.6 MHz for larger single-focus TUS. Can this phenomenon be understood as caused by the size of the ultrasound stimulation area, and the authors should comment on this statement? In addition, how can the authors prove that such high sound pressure will not cause damage to brain tissue or nerve cells?

Reply: The Reviewer presumably refers to the work in Ref. 20. Note, however, that there are many differences between Ref. 20 and our setup, not only the focal size. First, Ref. 20 uses awake mice versus anesthetized in our case. Also, they insert a glass rod into the brain whereas our approach is fully non-invasive. The frequency is also different - 0.6 MHz versus 3 MHz in our study. Notably, our setup is the first to isolate and compare the effect of the ultrasound stimulation area while keeping the ultrasound frequency constant.

To further clarify the holography-induced effects, we rephrased the text in the abstract as:

“Furthermore, holographic transcranial ultrasound stimulation (hTUS) allowed direct control of the stimulated volume and actively modulated local and mid-range network projections, effectively lowering the activation threshold by an order of magnitude.”

Further clarifications were also added in the second paragraph of the introduction:

“For example, single-element transducers produce different focal sizes at different frequencies due to wave diffraction. Thus, any attempt to evaluate the effect of frequency on stimulation with these transducers is inevitably confounded by the varying size of the stimulated area²⁶.”

In our previous report (Ref. 29), we have extensively studied any potential damages that can be caused by these levels of ultrasound pressure in the mouse brain. In particular, we performed real-time Calcium imaging during the sonication, functional connectivity analysis, and histology – all those confirming that repeated sonications of 150 ms duration up to 3 MPa did not cause damage. Clarification was added at the last paragraph of the Results section (TUS induces localized cortical activation) of the revised manuscript.

3) In Figures 3B and 3C, The authors mention that "In stark contrast, using hTUS for mouse cortex stimulation reveals a significant distributed excitation effect, achieving a robust activation response at only 1.2 MPa (Fig. 3B, C) with a triangular pattern where the foci centers are positioned at a 0.5 mm radius". Why did the authors set it to 0.5 mm here?

Reply: Due to the small focus size (Fig. 1D), the 0.5mm distance from the triangle's center ensures that each focus remains well defined without producing a merged pattern. In Fig. 4 we show results for different fields, including a triangle with a 1 mm radius. Clarification has now been added at the end of the first paragraph in Results section (TUS wavefront shaping enables dynamic steering and holographic enhancement of focal responses), as follows: "... a triangular pattern where the foci centers are positioned at a 0.5 mm radius, enabling a clear separation between each focal spot (see Fig. 1C).”

4) In Page 4, the authors mentioned that "At 1.1 MPa, pentagonal and triangular hTUS-triggered activation appear almost identical, nevertheless, pattern geometry does have an effect: for example, the larger triangular hTUS pattern (purple, 1 mm radius) excites the mouse cortex at 1.2 MPa but with a lower activation amplitude than its 0.5 mm radius counterpart.". From the results, it seems to have less to do with the pattern geometry (since pentagon and triangle have almost the same activation effect) and more to do with the distance between US firing points, right?

Reply: We apologize if the description was not clear. We rephrased the sentence as: “At 1.1 MPa, pentagonal and triangular hTUS-triggered activation appear almost identical, nevertheless, the size of the pattern does have an effect: for example, the larger triangular hTUS pattern (purple, 1 mm radius) ...”.

5) *In Page 4, the authors state that "TUS triggers Ca²⁺ responses only at powers above 0.1 W, whereas triangular and pentagonal hTUS stimulation modes achieve an activation area of about 2 mm² at around 0.1 W.". How was 0.1 W determined here and how was the 2 mm² calculated?*

Reply: We refer the Reviewer to the following description in the Methods: Fluorescence data processing: “The activated area is calculated using a flood-fill algorithm starting at the focal point for TUS or at the center of the hTUS pattern. The area is then calculated as the pixels fulfilling $\Delta F/F_0 > 0.5\%$ at any time between 0.46 s and 1s after the start of the US stimulus.” We now added the following clarification to the Methods section (Ultrasound excitation model): “Similarly, the acoustic power is obtained by integrating the axial acoustic intensity (I_z) along the xy plane at the focus.” An additional reference was added to the text mentioned by the Reviewer in the last paragraph of the Results section (TUS wavefront shaping enables dynamic steering and holographic enhancement of focal responses).

6) *In Figure 4C, why is the $\Delta F/F_0$ set to greater than 0.5%?*

Reply: We defined 0.5% as a threshold to measure the activated area. More details are answered above in question 5.

7) *How is the Total force of Figure 4F calculated?*

Reply: We now clarified this in the Methods (Ultrasound excitation model), as following:

“We first use linear acoustic simulations of the array based on the hydrophone measurements through a mouse skull using a spatial grid size of $40 \times 40 \times 40 \mu\text{m}^3$ and temporal step of $0.05 \mu\text{s}$. The high spatio-temporal sampling allows the calculation of the acoustic velocity and intensity. The radiation force density is calculated using the full vectorial expression found in Eq. (12) of Prieur & Sapozhnikov⁴⁵, which is later integrated along all three spatial dimensions to yield the total radiation force (see Eq. (20) in Supplementary Methods).”

8) *In the discussion, the authors claim that "Our method achieves precise focusing suitable for the dimensions of the mouse brain, closely approximating the relative focal dimensions used in targeting the human brain with TUS. Why do the authors say it is very close to the relative focus size used for TUS to target the human brain?"*

Reply: To clarify our claim concerning the relative volume of the ultrasound focus with respect to the total brain volume, we compiled the table below. Many other studies performed in mice use a similar focus volume as in humans. As a result, the relative size of the ultrasound focus with respect to the mouse brain volume is orders of magnitude larger (> 50%) than in our study, which hampers spatial specificity. Our system maintains a volume ratio of < 0.1%.

Species	Ref. #	Brain volume (cm ³)	Focus lateral (cm)	Focus axial (cm)	Focal volume (cm ³)	Volume ratio (%)
Human	49	1195	0.45	3.2	0.648	0.05

Mouse	This work	0.509	0.05	0.05	0.000125	0.02 – 0.06
Mouse	8	0.509	0.16	2.5	0.064	12.57
Mouse	9	0.509	N.A.	N.A.	N.A.	N.A.
Mouse	10	0.509	0.5 – 0.1	N.A.	N.A.	N.A.
Mouse	15	0.509	0.08	0.3	0.00192	0.38
Mouse	17	0.509	0.09	0.17	0.001377	0.27
Mouse	20	0.509	0.2	0.5	0.02	3.93
Mouse	21	0.509	0.44	2	0.3872	76.07
Mouse	23	0.509	0.5	N.A.	N.A.	N.A.
Mouse	24	0.509	0.3	0.75	0.0675	13.26
Mouse	25	0.509	0.57	N.A.	N.A.	N.A.
Mouse	26	0.509	0.135	N.A.	N.A.	N.A.
Mouse	34	0.509	0.14	0.8	0.01568	3.08
Mouse	35	0.509	N.A.	N.A.	N.A.	N.A.

Human brain volume from Cosgrove, K. P., Mazure, C. M. & Staley, J. K. Evolving Knowledge of Sex Differences in Brain Structure, Function, and Chemistry. *Biological Psychiatry* 62, 847–855 (2007).

Mouse brain volume from Badea, A., Ali-Sharief, A. A. & Johnson, G. A. Morphometric analysis of the C57BL/6J mouse brain. *NeuroImage* 37, 683–693 (2007).

We added this table as a Supplementary Table 3 and now mentioned it in the text referenced by the Reviewer at the beginning of the Discussion.

9) *The focal spot of ultrasonic stimulation used in the system is 250 μm , which is smaller than that used in most studies. Is the smaller the focal spot of ultrasonic stimulation the better?*

Reply: By using a smaller focus, we achieved a similar level of relative spatial specificity as what is commonly used in human studies. Therefore, our system is more suitable for the mouse brain dimensions, potentially also making it easier to cross-correlate the results.

10) *In this paper, the system uses a 3 MHz probe. Because the human skull is relatively thick, in order to improve the penetration depth and reduce the impact of scattering and reflection, the ultrasonic frequency is usually used between 200 kHz and 1.5 MHz. Why 3 MHz ultrasound used here? In addition, does the frequency of ultrasound affect the effect of regulation?*

Reply: Our frequency is higher than what is normally used in human applications because we aim to achieve a similar (high) level of spatial specificity in mice. We do not intend to directly translate our system to humans. The main purpose is to show that the size of the stimulated regions makes a big difference. This aspect has been consistently overlooked in all previous studies performed in small rodents because it is impossible to control for the focus size using single element ultrasound transducers of different frequencies. Therefore, frequency plays a role, but less than previously thought. Further research should be carried out in the future where the role of both frequency and stimulation focus size is studied independently.

Minor concerns:

1) *In the Figure caption of Figure 1, "... the a 512-element spherically focused array".*

Reply: We corrected this issue in the revised version.

2) *In the discussion, the "result" in the sentence "While efforts are being made to continue perfecting these computational methods, further experimental work is needed to validate our results and accurately measure radiation forces in the brain" should be "results".*

Reply: We corrected this issue in the revised version.

3) *In the supplementary document, Page 2, "In doing that, the number of elements used to generate a focus change from 512 for single focus to $512/3 \sim 170$ for a triangle and $512/5 \sim 102$." should add "for a pentagon" at the end of the sentence.*

Reply: We corrected this issue in the revised version.

4) *In the supplementary document, Page 2, The "LISTNUMLISTNUM" follows formula (1) should be deleted.*

Reply: We corrected this issue in the revised version.

Reviewer #3 (Report for the authors (Required)):

The study by Estrada et al. describes the application of a holographic transcranial ultrasound stimulation (hTUS) technique to improve neuromodulation through engaging multiple brain circuits in anesthetized mice. The authors delivered 3MHz US using a 512 channels spherical matrix array to focus to ~250um X 500um, and validated focal activation by measuring cortical Ca responses using GCamp6f. A key observation is that sonicating multiple foci lowered the threshold of Ca activation. Using a USEM model, the study identified a unified Ca activation threshold of 0.15mN. Furthermore, the study attempted to evaluate network versus cellular effects using a cortical neural network model containing individual neurons modelled biophysically, and demonstrated thermal and voltage effects. Overall, this study presents a good solution to shape US focus to meet the challenge of the small head in mice. However, the impact of the work, especially, the translational impact is limited. The observation that sonicating multiple area, collectively a largely area, reduces activation threshold is interesting. The modeling attempt however cannot offer concrete mechanistic insights.

Reply: We thank the Reviewer for their assessment and hope they will find our revisions satisfactory.

Major Comments:

1. It is unclear about the significance of the work. The introduction lacks detailed discussion on the existing gaps to justify the need for the current research. Although this tool enables better focusing for mice, 3 MHz US does not pass through the human cranium as well as sub-MHz frequencies. Could the authors comment more on how mechanistic results derived from these methods may differ from the lower frequencies typically used in human studies? For example, radiation force is more likely a mechanism at higher frequencies given its relationship with absorption.

Reply: We thank the Reviewer for their constructive criticism. Based on the Reviewer's concern in question 8 to "address sonication area as a potential confounding factor", we understand that we did not properly communicate the significance of our work. In fact, sonication area/volume is the main driving parameter of our observed effects, and holography represents a unique solution to control for this volume while keeping other stimulation parameters (such as carrier frequency) fixed. We rephrased the text in the abstract to make this point clearer, i.e.:

"Furthermore, holographic transcranial ultrasound stimulation (hTUS) allowed direct control of the stimulated volume and actively modulated local and mid-range network projections, effectively lowering the activation threshold by an order of magnitude."

The second paragraph of the introduction was amended as follows:

"For example, single-element transducers produce different focal sizes at different frequencies due to wave diffraction. Thus, any attempt to evaluate the effect of frequency on stimulation with these transducers is inevitably confounded by the varying size of the stimulated area²⁶."

We choose 3 MHz to mimic the stimulated volume relative to total volume that is currently achieved in humans (see new Supplementary Table 3 provided in response to question 3 by Reviewer 1). Thanks to our small focal size better fitting the mouse brain dimensions, we can observe the critical effect of the focal size on evoked responses due to network excitability. This effect has been overlooked in the literature where frequency alone has been

presented as a crucial parameter affecting ultrasound neuromodulation effects, without considering its impact on focal volume. We therefore believe that our finding that hTUS (i.e., larger focal volumes) can more efficiently excite brain networks is not specific to 3 MHz. We added the following paragraph in the discussion section to make our point clearer:

“Although lower frequencies capable of penetrating the human skull have been used in mice²⁰, this approach is not capable of generating tightly focused responses. Focal excitation at similar frequencies to ours (2 MHz) combined with cellular resolution imaging⁴⁷ adds a complementary view by showing the dependence of TUS on I_{SPTA} . It remains to be seen how our observations translate to sub - MHz range used to traverse the human skull.”

The Reviewer’s question whether radiation force as a unifying mechanism is translatable to the human brain is a very relevant one. Our results demonstrate that previous evidence must be reassessed because even the same frequency (regardless of the mechanism) can produce different activation thresholds. We refrain from making claims in the manuscript concerning the exact mechanism that can be translated to humans.

2. The study claims precise focusing and steering capabilities but does not thoroughly discuss the limitations, particularly steering accuracy. Although the focal size (~250 μm laterally, 500 μm axially) is mentioned, it is not compared to focal sizes in other TUS studies in mice. Additionally, while the study states that the focal size to brain region ratio is similar to human studies, it lacks a detailed quantitative comparison between human and mouse brain targeting ratios.

Reply: We added new Supplementary Table 3 to answer the question regarding spatial specificity compared to other studies and referred to it in the first paragraph of the Discussion (page 6). Note that our manuscript focuses on demonstrating the critical role of network interactions in transcranial ultrasound neuromodulation and how it could be harnessed using holography. The steering accuracy can be inferred by standard wave diffraction theory and Supplementary Figs. 1–3, however, we believe that a thorough technical discussion and experimental validation of the steering accuracy is out of the scope of our study.

3. Lacks control data that could serve as a baseline for comparison with the TUS-treated groups.

Reply: We thank the Reviewer for sharing their concern. Due to the small focus size we used a within-subject control design, comparing TUS-treated brain hemisphere with untreated hemisphere from the same subject. This approach offers several key advantages: It minimizes variability by controlling for individual differences, it increases statistical power since each subject serves as their own control and it reduces ethical issues by avoiding unnecessary sham treatments. Based on our control findings, we believe a sham control would not add significant new information. In addition, our setup can deliver TUS and hTUS in the same mouse without having to change any experimental conditions besides the configuration of the electronics. If one would like to perform the same test with different single element transducers, changing the transducers while aiming at the same region would introduce uncertainties that are actually avoided by our study design.

4. The study measures responses in the auditory region to identify potential auditory confounds. However, without detailed methodology, control experiments, and quantitative analysis, it is difficult to conclusively differentiate between direct neural activation and indirect auditory effects.

Reply: The auditory regions were included in the field of view and analyzed (orange time traces in Figs. 2A–D, 3B–C), revealing no clear activation. Moreover, our focal Calcium responses are spatiotemporally locked to the ultrasound stimulus. For instance, it is clearly visible in Fig. 3A that the responses are mostly restricted to the focus even if we change its position. Based on these observations we are confident that Calcium responses indicate direct neural activations rather than indirect responses mediated by an auditory pathway.

5. The study uses the area under the curve (AUC) to analyze Ca²⁺ responses in different brain regions following TUS, comparing the target site to auditory and off-target regions. While AUC measures overall activity, it can obscure temporal dynamics and peak response characteristics. A more detailed analysis of peak amplitude, response latency, and temporal patterns would help differentiate direct TUS effects from confounds.

Reply: We thank the Reviewer for sharing their concern. As pointed out in our response to question 4, our calcium responses are spatiotemporally locked to the ultrasound stimuli and can be reliably steered with the acoustic focus, indicating a robust and direct mechanism rather than a confound. We appended the following text at the end of the last paragraph of the Results section (TUS induces localized cortical activation) and added a related figure as supplementary Fig. 8. All data behind this plot is available as supplementary data for further reference.

“Based on the peak fluorescence data (see Supplementary Fig. 8B), fluorescence in the auditory region does not significantly change with TUS pressure ($p = 0.758$). Additionally, off-target peak fluorescence remains consistently below 0.5% for pressures exceeding 2.3 MPa, in agreement with a shorter time-to-peak (see Supplementary Fig. 8C) at the stimulated region”

Supplementary Figure 8: Extended single focus activation statistics. A) Change in the area under the time trace at points indicated by the labels as function of the peak pressure between 0.45 and 2 s from the US sonication start. One-way ANOVA analysis ($n = 3$) performed on three brain regions indicated by labels: Stimulated focus with ($F = 5.19$, $p = 0.009$) and without FTT correction ($F=5.51$, $p = 0.007$), out of focus ($F=2.36$, $p = 0.104$), and auditory region ($F=0.68$, $p = 0.650$). B) Peak fluorescence (between 0.45 and 2 s from the US sonication start) as a function of the peak pressure. One-way ANOVA analysis ($n = 3$) at the stimulated focus with ($F=5.25$, $p = 0.009$) and without FTT correction ($F=5.44$, $p = 0.008$), out of focus ($F=5.16$, $p = 0.009$), and auditory region ($F=0.51$, $p = 0.758$). C) Time to peak as a function of the peak pressure.

6. Figure 3 mentions minor crosstalk observed in the time traces but does not provide a quantitative analysis of this crosstalk.

Reply: We thank the Reviewer for pointing this out. In response, we added the following text to the Results section (TUS wavefront shaping enables dynamic steering and holographic enhancement of focal responses):

“Although a minor crosstalk of up to 40% is observed in the time traces”. The caption of Fig. 3A was amended as follows: “Percentages on the time traces represent the relative amplitude of the fluorescence with respect to the point of stimulation (100 %).”

7. The assumptions made in developing the cortical network model and their potential limitations are not fully discussed.

Reply: We added an in-depth discussion paragraph (Discussion section, 2nd paragraph) on the assumptions made in the implementation of our sCNM model, the reasons guiding our modeling choices, and the overall limitations of the model.

We also added further justifications for our selection of model parameters in the Methods section (Simplified cortical network model (sCNM)).

8. While the network model in Figure 5 is a promising potential physiological mechanism, it is difficult to interpret the experimental differences between TUS and hTUS without addressing sonication area as a potential confounding factor. Could these differences be explained by a higher volume of sonicated tissue in the hTUS condition? It would be interesting to compare hTUS to a larger single focal spot TUS with similar focal volume to the hTUS condition (potentially by patterning multiple overlapping hTUS spots). This may help decouple the impact of sonicated tissue volume vs local network projections in influencing observed changes in activation threshold.

Reply: We thank the Reviewer for their feedback. As previously mentioned, sonication area is not a confounding factor, but rather the main driving parameter behind the observed effects and our variable of interest in the study, which we control by holographic patterning.

While assessing the impact of focal size in a single spot arrangement would be ideal, patterning multiple overlapping TUS spots would confound the axial dimension. This is because, already at 250 μm separation radius, we obtain major stretching and distortion of the axial dimension (see lower panel in the figure below), which will confound any comparison.

Given these technical limitations, we are constrained to evaluating the impact of the sonicated volume using non-overlapped TUS spots. In this context, we implemented our network model with non-overlapping nodes to best replicate our empirical configuration. Nonetheless, because we model the acoustic input as a discrete set of pressure amplitudes at each node, we are theoretically not constrained to any specific geometrical arrangement. Therefore, our modeling results (response enhancement by multi-nodal stimulation) can also

be interpreted in a context where the nodes are directly overlapping, which would effectively correspond to a single, but larger TUS focus.

To address the Reviewer's more general concern about the lack of mechanistic insights of our network model, we now also clarified our analysis of model predictions. Specifically, we analyzed TUS-evoked effects in an isolated network node (revised Fig. 5C) to clarify the impact of each transmembrane current on membrane dynamics and spiking activity during a TUS stimulus, in the absence of network contributions. To further clarify this causal impact, we also report the cumulative charge injected by each transmembrane current over relevant time windows. We then analyze TUS-evoked effects in the full network model (revised Fig. 5D), focusing this time on the impact of synaptic currents on membrane dynamics and spiking activity. Here again, we report cumulative injected charges (revised Fig. 5E) to illustrate the causal role of synaptic currents in mediating response differentiation between TUS and hTUS.

9. Why did the study use TUS parameters that induce tissue heating (i.e., continuous wave US)? How might thermal effects be decoupled from ultrasound-induced mechanical responses in this study?

Reply: We did not specifically choose the TUS parameters to induce tissue heating, but rather because short (100 to 200 ms) continuous-wave TUS stimuli have been shown to elicit robust and dose-dependent calcium responses (see Ref. 20). On the other hand, tissue heating helps us establish the spatiotemporal relation between the stimulus and the Ca²⁺ activation. We investigated this thermal impact by including temperature-dependent effects in both models (see Results, US excitation model beyond peak pressure and simplified cortical network model). We found that thermal effects have a limited impact on our results.

10. It would be helpful to discuss and compare TUS and hTUS axial focal zones as well, as a key strength of this work is achieving precise TUS focality.

Reply: We added the figure below as Supplementary Fig. 3. As can be seen, holographic patterning has little impact on the axial focality achieved by our system.

Supplementary Figure 3. TUS and hTUS pressure field measurements. Free-field hydrophone scans for TUS and hTUS pressure fields at 3 MHz. Lateral (top row) and axial (bottom row) FWHM are shown by labels. Scale bars correspond to 500 μm .

11. *Why might hTUS be less precisely focused than steered TUS? Is it because hTUS creates more tissue heating?*

Reply: If the Reviewer refers to Fig. 1C (bottom), the in-vivo transcranial focus monitoring looks noisier for hTUS than for steered TUS because the signal-to-noise ratio is lower for hTUS due to its lower peak pressure. While steered TUS uses all the 512 elements of the array, hTUS uses a fraction of the elements for each focus. Thus, steered TUS actually creates more tissue heating than hTUS, since it generates a larger peak pressure. However, as shown in Supplementary Methods (Holographic focused ultrasound) and Supplementary Figs. 1–3, the maximum sidelobe amplitude is 20% of peak pressure for the pentagon pattern, implying that the focusing precision is only slightly different.

Minor Comments:

1. *The title is generally accurate but could benefit from slight adjustments to better reflect the specific enhancements demonstrated, the extent of mechanistic understanding, and the model used.*

Reply: We thank the Reviewer for their suggestion. The title has been amended as “Holographic transcranial ultrasound neuromodulation enhances stimulation efficacy by cooperatively recruiting distributed brain circuits”. In lieu of other revisions and clarifications, we believe the new title better highlights the significance of our work, particularly in context of the impact of sonication area.

2. *Duty cycle, pulse repetition frequency, pulse duration, and ISPPA/ISPTA are not listed.*

Reply: I_{SPTA} has been omitted since we use continuous-wave pulses (duty cycle = 100%, pulse repetition frequency = 0 Hz). We added the information at the first paragraph of the Results section (TUS induces localized cortical activation) as “The right hemisphere was sonicated 20 times at 10 s intervals with continuous 150 ms US pulses ($20 < I_{SPPA} < 131 \text{ W/cm}^2$).”

3. *Within figure 1, the ultrasound waveform used should be added.*

Reply: We added the transcranially measured waveform as Fig. 1E.

4. *Figure 1A lacks detailed labels and annotations (red / blue colors) that explain the key components.*

Reply: We added the labels “Emitted ultrasound” in blue and “Laser-generated US” in red. The caption includes the text “spherically focused array for hTUS delivery (blue) navigated with volumetric optoacoustic tomography (red) performed with the same array.” regarding Fig. 1A.

5. *The in vivo characterization shown in Figure 1C should specify the exact brain region, or at least within the figure caption.*

Reply: Thank you for this suggestion. We added “near the somatosensory area” to the caption of Fig. 1.

6. *The study measures Ca²⁺ signals in various regions of the mouse brain, including the target stimulation site and potential auditory regions, to assess neural activation induced by TUS (Figure 2). However, the specific regions of interest for these measurements are not clearly defined, dorso / lateral / etc, making it challenging to interpret the spatial specificity and relevance of the observed responses.*

Reply: We thank the Reviewer for pointing this out. Figs. 2B, 3A, and 3C have a standard mouse brain atlas overlaid to facilitate the interpretation. In addition, colored circles indicate the regions of interest where the time traces and statistics are obtained. We added new labels to help the reader identify brain regions in Figs. 2B, 3A, and 3C; as well as the following text at the end of the first paragraph at the Results section (TUS induces localized cortical activation): “the point of TUS delivery (red spot) at the primary somatosensory cortex – lower limb region (SSp-II, anterior-posterior ~ -1.2mm from bregma, medial-lateral ~-1.6 mm).”

We also added to the caption of Fig. 2 “The sonicated point is red (1.2 mm posterior to bregma, 1.6 right to midline), the out-of-focus is blue (1.2 mm anterior to focus point)” “Anatomical labels in B) i: Primary motor area, ii: Primary somatosensory area (lower limb), iii: Primary visual area, iv: Primary auditory area. See Supplementary Fig. 7 for a more complete mouse brain atlas reference.”.

7. *The time course of local transmembrane currents and membrane potentials (Figure 5B) is presented, but the traces are densely packed and difficult to read.*

Reply: We have unpacked the membrane current traces in Fig. 5 to facilitate visual inspection and interpretation.

8. *The methods lack detailed information on the calibration and validation of the ultrasound array to ensure accurate and consistent pressure fields.*

Reply: We added the following paragraph in the Methods (hTUS characterization):

“Characterization of the TUS and hTUS pressure field is shown in Supplementary Fig. 3. Measurement of the transcranial pressure using an excised mouse skull was performed for different voltages and frequencies (see Fig. 1E and Ref. ²⁹) and was used to calibrate the ultrasound simulation.”

We also added Supplementary Fig. 3 with more detailed measured ultrasound fields. The array and its effects on the mouse brain have been extensively characterized elsewhere (see Refs. 16, 28, and 29). To the best of our knowledge, our work is the first to show in-vivo transcranial pressure distribution measurements in the mouse brain (see Fig. 1C).

9. *The discussion sometimes overgeneralizes the results without adequately considering the limitations of the study.*

Reply: We apologize if any of our statements have been perceived as overgeneralization. As clarified in our answer to question 1, our results demonstrate the importance of the pressure field distribution on evoked effects independently of acoustic frequency, which suggests that ultrasound waves actively interact with the network structure of the brain. We believe that our discussion faithfully conveys this message and hope that our clarifications and revisions have mitigated the Reviewer's concerns about overgeneralization. In addition, the discussion now includes the following elaborated text explaining limitations of our study:

“First, the nature and strength of TUS' direct effects on cellular membrane dynamics and resulting neural activation are currently unclear^{36–40}. Our sCNM model therefore assumes a simplistic mechano-electrical transduction whereby TUS acoustic pressure is directly converted into a depolarizing membrane current. This mechanism-agnostic strategy captures the pressure dependency of TUS-evoked responses through a single parameter, which can be tuned to replicating our empirical excitation thresholds. Second, the mechanisms by which TUS-evoked temperature variations affect neural activity remain to be elucidated. We therefore pursued various strategies to incorporate temperature sensitivity in our sCNM model based on established literature^{32,41} as well as recent empirical evidence in mouse cortical and striatal neurons³³ (see Methods). Simulations in our thermally sensitive model indicate that temperature variations induced in our experiments have minimal impact on neural activity, and that voltage variations and spiking activity itself play a much more predominant role in regulating neural excitability (Fig. 5C). Finally, our sCNM model adopts a simple network architecture consisting of excitatory-only point-neuron models in a fully connected network. While this architecture does not represent the full complexity of cortical circuits, it nonetheless captures essential features of their organization that are relevant in the context of our study. For instance, there is significant evidence that mid-range synaptic projections – such as those occurring between TUS-hotspots – are primarily excitatory⁴². Hence, despite several simplifications, our sCNM provides valuable insights on the network mechanisms enabling enhanced cortical responses to hTUS.”

“It remains to be seen how our observations translate to sub - MHz range used to traverse the skull in humans.”

“While efforts continue to refine these calculation methods⁴⁶, further experimental work is needed to validate our results and accurately measure the radiation force in brain tissue”

In this paper, the authors studied the application of holographic ultrasound in neuromodulation, and proved that transcranial ultrasound can achieve high-precision and dynamically controllable nerve stimulation, and holographic transcranial ultrasound stimulation (hTUS) can significantly reduce the neural activation threshold. The experimental results show that hTUS can effectively modulate local and mid-range neural network projection, revealing the potential of complex interaction mechanism between ultrasound and neural tissue. The results of the paper are of great significance to the progress of non-invasive brain network regulation technology, and provide new methods and perspectives for future neuroscience research and treatment of neurological diseases. However, there are still some concerns need to be solved.

Major concerns:

- 1) In the fluorescence image on the left of Figure 2B, why is the fluorescence intensity strong in the lower left corner? And after a period of ultrasonic stimulation, the intensity went away again?
- 2) In Figure 2C, the authors used sound pressures of 2.5 MPa or higher to show a more pronounced rise in Ca^{2+} response, significantly higher than the response threshold recently observed at 0.6 MHz for larger single-focus TUS. Can this phenomenon be understood as caused by the size of the ultrasound stimulation area, and the authors should comment on this statement? In addition, how can the authors prove that such high sound pressure will not cause damage to brain tissue or nerve cells?
- 3) In Figures 3B and 3C, The authors mention that "In stark contrast, using hTUS for mouse cortex stimulation reveals a significant distributed excitation effect, achieving a

robust activation response at only 1.2 MPa (Fig. 3B, C) with a triangular pattern where the foci centers are positioned at a 0.5 mm radius". Why did the authors set it to 0.5 mm here?

4) In Page 4, the authors mentioned that "At 1.1 MPa, pentagonal and triangular hTUS-triggered activation appear almost identical, nevertheless, pattern geometry does have an effect: for example, the larger triangular hTUS pattern (purple, 1 mm radius) excites the mouse cortex at 1.2 MPa but with a lower activation amplitude than its 0.5 mm radius counterpart.". From the results, it seems to have less to do with the pattern geometry (since pentagon and triangle have almost the same activation effect) and more to do with the distance between US firing points, right?

5) In Page 4, the authors state that "TUS triggers Ca^{2+} responses only at powers above 0.1 W, whereas triangular and pentagonal hTUS stimulation modes achieve an activation area of about 2 mm² at around 0.1 W.". How was 0.1 W determined here and how was the 2 mm² calculated?

6) In Figure 4C, why is the $\Delta F/F_0$ set to greater than 0.5%?

7) How is the Total force of Figure 4F calculated?

8) In the discussion, the authors claim that "Our method achieves precise focusing suitable for the dimensions of the mouse brain, closely approximating the relative focal dimensions used in targeting the human brain with TUS. Why do the authors say it is very close to the relative focus size used for TUS to target the human brain?

9) The focal spot of ultrasonic stimulation used in the system is 250 μm , which is smaller than that used in most studies. Is the smaller the focal spot of ultrasonic

stimulation the better?

10) In this paper, the system uses a 3 MHz probe. Because the human skull is relatively thick, in order to improve the penetration depth and reduce the impact of scattering and reflection, the ultrasonic frequency is usually used between 200 kHz and 1.5 MHz. Why 3 MHz ultrasound used here? In addition, does the frequency of ultrasound affect the effect of regulation?

Minor concerns:

- 1) In the Figure caption of Figure 1, "... the a 512-element spherically focused array".
- 2) In the discussion, the "result" in the sentence "While efforts are being made to continue perfecting these computational methods, further experimental work is needed to validate our results and accurately measure radiation forces in the brain" should be "results".
- 3) In the supplementary document, Page 2, "In doing that, the number of elements used to generate a focus change from 512 for single focus to $512/3 \sim 170$ for a triangle and $512/5 \sim 102$." should add "for a pentagon" at the end of the sentence.
- 4) In the supplementary document, Page 2, The "*LISTNUMLISTNUM*" follows formula (1) should be deleted.

The authors have essentially addressed the issues that I raised in the revised manuscript.

However, there are still a few minor issues that require further clarification.

1. Figure 1A clearly illustrates the optoacoustic setup, but it appears that no results related to optoacoustic imaging are presented in the manuscript. Although the authors refer to previous studies demonstrating the use of optoacoustic tomography for precise localization of the stimulation spot in the mouse brain, the absence of any direct optoacoustic results in this study raises questions about the relevance of this setup to the current work. I would recommend either: a) Including optoacoustic imaging results to demonstrate how this technique was used to localize the stimulation spot, thereby enhancing the completeness of the study; or b) Removing the optoacoustic setup from Figure 1 if it was not utilized in the current experiments, to avoid confusion and maintain focus on the primary methods and results of this study.

This clarification would help to streamline the presentation of the manuscript and ensure that all components of the experimental setup are relevant to the findings reported.

2. The authors stated, “Although minor crosstalk up to 40% is observed in the time traces, the peak activation occurs at the targeted location with a pressure of 2.8 MPa.”

However, is it appropriate to characterize 40% crosstalk as “minor” given that 40% is quite a significant value? It is recommended to remove the word “minor.”